# Energy drinks in Tamale: Understanding youth perceptions, consumption patterns, and related factors

**Williams Kobik** [1]☯*, **Paul Armah Aryee**[2]☯

**1** Department of Health and Nutrition, Allied Relief, Tamale, Northern Region, Ghana, **2** Department of Dietetics, University for Development Studies, Tamale, Northern Region, Ghana

☯ These authors contributed equally to this work.
* williamskobik@gmail.com

**Data Availability Statement:** All the data is available on figshare (10.6084/m9.figshare. 24716805).

**Funding:** The authors received no specific funding for this work.

## Abstract

Energy drinks (EDs) have become a popular choice for young people seeking physical and cognitive boosts, with ingredients such as caffeine, taurine, and B vitamins aimed at improving academic, athletic, and alertness levels. However, the popularity of these drinks is also driven by low prices, taste, brand loyalty, and gendered marketing, with boys being more likely to consume them. Despite the supposed benefits, EDs have been associated with high-risk behaviours, deaths, and adverse health effects, especially those related to cardiovascular risk. Meanwhile, in Ghana, the use of EDs is on the rise. Hence, this study aimed to examine the prevalence and consumption patterns, perceptions, and factors associated with ED consumption among the youth of the Tamale Metropolis. The study was cross-sectional, consisting of 541 participants. The group consisted of 340 males and 201 females, between the ages of 15 and 45. A questionnaire was utilized to obtain data on the respondents' consumption patterns and perceptions of EDs, as well as their socio-demographic characteristics. The results of the study indicated that a large percentage of the respondents, 98.7%, had consumed EDs before, while 78.7% currently consume them. Respondents believed that EDs provided additional energy (81.00%) and reduced stress (62.30%). However, they also perceived side effects such as insomnia (60.60%) and restlessness (51.40%). Also, the highest proportion of respondents (83.4%) had poor knowledge of EDs. They were unaware of the classification of EDs and their ingredients, side effects, and benefits. Age, marital status, level of education, work intensity, EDs served at gatherings, and knowledge of EDs was significantly associated with ED consumption (p < .05). Consumption was higher among those aged 26 to 35 years, singles, individuals with no formal education, and those with high work intensity. The high consumption was also associated with low knowledge levels. It is recommended that public health and nutrition professionals engage in further advocacy efforts to improve the youth's knowledge and perception of EDs in a positive manner. In addition, lawmakers should use legislation to influence consumption rates and safeguard the health of consumers.

**Competing interests:** The authors have declared that no competing interests exist.

## Introduction

Throughout history, humans have sought physical and cognitive enhancements to tackle challenging tasks [1]. Conveniently, Energy Drink (ED) marketing strategies are centered around physical boosts and improved cognitive performance [2]. Thus, there has been a significant increase in consumption among young people [3]. Ingredients commonly found in EDs include caffeine, guarana, ginseng, taurine, glucose, L-carnitine, glucuronolactone, and B vitamins [4]. Despite their perceived benefits, factors such as low cost, taste, brand loyalty, and gendered marketing have contributed to their popularity. Notably, marketing campaigns geared towards masculinity have resulted in fewer girls consuming EDs than boys [5, 6]. Over the course of history, sports have served as a domain where masculinity is learned, demonstrated, and perpetuated [7]. Indeed, since their inception in 1987, the marketing of these products has heavily depended on visuals linked with extreme sports and risk-taking behaviours [8, 9].

The ED market is one of the fastest-growing segments of the non-alcoholic beverage industry. Despite the notoriety of Eds for being high in sugar and caffeine, their dangers have been largely overlooked. Meanwhile, the European Union has stated that there is no evidence to support the argument that EDs pose a degree of toxicity to the extent of adverse health outcomes. However, case reports from around the world suggest otherwise [10–12].

EDs have not only been associated with negative health effects and death but have also been linked to high-risk substance use and social deviance such as vandalism and reckless driving, resulting in accidents [4, 13–15]. Acute effects include sudden cardiac death, high blood pressure, and endothelial dysfunction, which can affect young people and those with underlying cardiovascular conditions. The usual ingredients found in EDs, including guarana, ginseng, taurine, and caffeine, have been associated with a range of adverse health outcomes [16]. Also consuming excessive sugar commonly found in EDs, contributes to overnutrition [17].

In the specific context of Ghana, a nation within a sub-region grappling with an uptrend in diet-related diseases [18], there has been a discernible uptick in ED consumption [19]. A survey revealed that 62% of student-athletes consume at least one can of ED per week [20]. This may be the result of the unregulated sale and use of EDs in Ghana. Compared to some developed countries such as France, Denmark, and Norway, EDs with high levels of caffeine and taurine have been banned, such restrictions are absent in Ghana. In some other countries, it is required that EDs be sold as medicinal products or with warning labels [19, 20].

This study builds upon existing research on ED consumption in Ghana, expanding beyond previous investigations that predominantly focused on specific cohorts like students, student-athletes, and drivers [19–24]. However, they have left a gap in our understanding of ED consumption among the general youth population. This study addresses this gap by examining ED consumption among a representative sample of Ghanaian youth.

The overall aim of this study was to evaluate the youth's perceptions, consumption habits, and factors linked with ED consumption in the Tamale Metropolis.

This study enhances our understanding of ED consumption habits among the youth in the Tamale Metropolis. The findings can raise awareness, prompt further research in this domain, and influence policy and public health initiatives.

## Methods

### Study design

The study design was a descriptive cross-sectional. This approach was primarily selected for its suitability in capturing a snapshot of the population's characteristics and behaviours at a

specific point in time. This design was deemed appropriate for the research objectives. It allowed for a comprehensive examination of ED consumption patterns among the study participants within the defined timeframe. The reasons for utilizing a descriptive cross-sectional design lie in its efficiency in providing a quick and cost-effective overview of the prevalence and distribution of the studied variables. This approach facilitates the identification of trends, making it particularly valuable for investigating the current landscape of ED consumption in the target population. Additionally, the design enables the simultaneous assessment of multiple factors, contributing to a holistic understanding of ED consumption within the constraints of a singular data collection period.

## Study area

With a population of about 360,579, the Tamale Metropolis is located in the Northern Region. Of this, 80.8% reside in urban areas. The metropolis spans 289.58 square miles [25]. EDs are readily available through convenience shops and street vendors, and the area has a fast-growing business landscape where the trading of EDs thrives. Below is a map of the metropolis extracted from Google Maps (Fig 1).

## Study site

The data were collected in the central business district of the metropolis.

## Study population

The focus of this research was on individuals aged 15 to 45 years old. The demographic aged 15 to 45 comprises approximately 63.3% of the population engaged in economic activities which demonstrates financial capability for purchasing EDs [25]. This group, primarily consisting of teenagers and young adults, holds specific beliefs about the perceived advantages of these beverages, influencing their decision to consume them. Given that individuals in this age range are actively exposed to marketing and advertising for EDs and are still in the process of developing their health habits, understanding their consumption patterns is crucial. The engagement of adolescents and young adults in sports, extracurricular activities, and social events further underscores the relevance of comprehending the factors influencing their ED

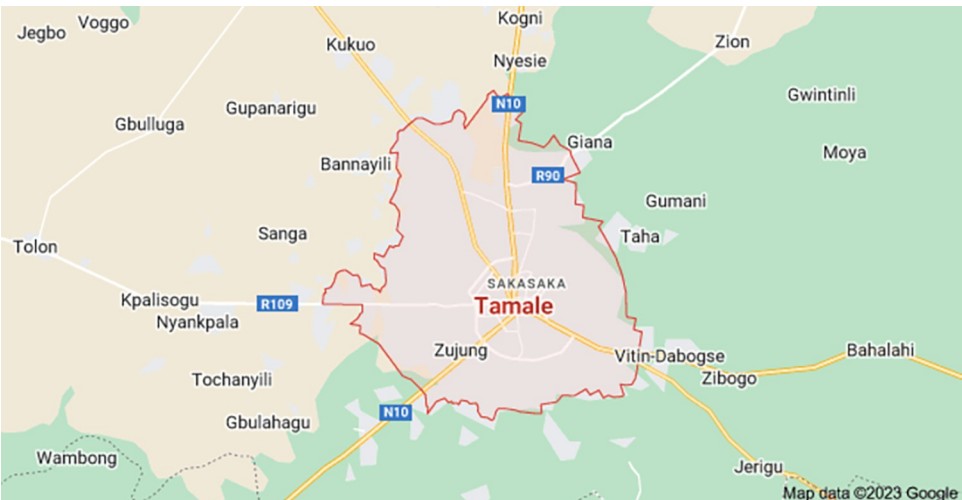

**Fig 1. Map of Tamale (Coordinates: 9.404601, -0.842389).**

consumption. This knowledge is pivotal for promoting healthy behaviors and mitigating potential health risks associated with their perceptions of ED benefits.

### Inclusion and exclusion criteria

The participants for this study were required to fall within the age range of 15 to 45 years and also be permanent residents of the Tamale metropolis. To verify residency, respondents were asked if they had lived in the metropolis for a minimum of five years. These criteria were set to ensure that the data collected was representative of the local population within this age bracket and who have had adequate exposure to the city's environment and social dynamics.

### Sample size determination

The total population of youth (n) for this study, 383 was determined using Cochran's formula. The confidence interval was set at 95% with the z-score (t) being 1.96. The margin of error (m) was 0.05 and the prevalence of ED consumption (p) was estimated to be 46.7% [21].

### Sampling technique

The Tamale Metropolis was purposively chosen as the study location, and respondents were selected for convenience. Specifically, study participants were chosen based on their presence in the central business district of the metropolis. Although convenience sampling was used to select participants, the process depended on the established inclusion and exclusion criteria.

### Data collection and quality management

The study collected data on individual respondents through semi-structured interviews. The questionnaire was pretested and then administered via face-to-face interviews with the aid of a Computer Assisted Personal Interview (CAPI) device. The questionnaire was divided into four sections to cover demographic information, anthropometry, and perception of EDs. In terms of demographic information, the study gathered data on respondents' age, gender, marital status, ethnicity, religion, employment status, work intensity, educational level, and smoking and drinking behaviours. Anthropometric measurements included height and weight parameters. The perception section of the questionnaire probed respondents' knowledge of EDs, exploring factors such as awareness, attitudes, and beliefs. It sought to probe respondents' understanding of EDs, including their ingredients, brands, benefits, and adverse effects. Based on the responses, a score was generated, and respondents' knowledge of EDs was eventually categorized as "poor," "good," or "excellent." The response scales and metrics used in the perception section were Likert, multi-choice, and frequency scales. The study employed a combination of closed-ended questions with predefined response options, as well as open-ended questions to capture a comprehensive understanding of respondents' perceptions regarding EDs. Additionally, the data was collected by enumerators who received a one-day training on questionnaire administration and translation into the local dialect.

### Data analysis

The collected data was transferred from the Kobo toolbox on mobile phones to Excel for cleaning and then to IBM Statistical Package for Social Sciences (SPSS version 21) for analysis. The consumption patterns of EDs were analyzed using frequencies and percentages. For continuous data, correlations, means, and standard deviations were used. The relationship between exposure variables such as demographics, perceptions and consumption patterns was examined using Chi-square tests.

The data gathered explored respondents' socio-demographic characteristics, knowledge, attitudes, and practices regarding EDs. Analysis was done using various statistical tests, including T-tests, chi-square tests, Pearson's r correlation, and logistic regression, with a significance level of 0.05.

### Ethical considerations

The study followed ethical guidelines and obtained ethical approval from the Committee on Human Research and Publication Ethics (CHRPE) with Ref No: CHRPE/AP/256/22. Consent was obtained orally. The decision to obtain oral consent in this research on ED consumption was meticulously considered, factoring in ethical principles and practical considerations specific to the study. Participants were easily accessible in the central business district of the metropolis, facilitating efficient oral consent acquisition during brief interactions. The research, focusing on surveying participants about their ED consumption habits, presented minimal risk to the health of individuals. Given the impracticality of obtaining written consent from a sizable participant pool, oral consent emerged as a more practical alternative, streamlining the consent process. Before data collection, informed consent was obtained from all participants, emphasizing their voluntary participation and the confidentiality of their information. Participants were provided with clear explanations regarding the purpose, procedures, and potential risks and benefits associated with the study. They were also assured of their right to withdraw from the study at any point without repercussions. To uphold privacy and confidentiality, personal identifiers were removed from the collected data, and data storage followed secure protocols. Only authorized researchers had access to the data, maintaining strict confidentiality throughout the study.

## Results

### Socio-demographic characteristics of youth

The respondents had a mean age of 25.55 ($SD$ = 8.11) years, with the majority (59.7%) falling in the 15–25 age range. Most respondents were male (62.8%), single (70.6%), practicing Islam (66.0%), and from the Dagomba ethnic group (55.1%). Around 44% of respondents were students, and a significant proportion had formal education. The majority (65.2%) engaged in labor-intensive work. An independent samples T-test revealed significant age differences between males and females ($t$ (366.66) = -2.04, $p$ < .001), with females ($M$ = 36.51, $SD$ = 8.92) being older than males ($M$ = 24.98, $SD$ = 7.55). However, there were no significant BMI differences between genders ($t$ (539) = 1.623, $p$ = .105). The chi-square test examined associations between gender and general characteristics. Marital status did not differ significantly by gender ($X^2$(4, N = 541) = 4.67, $p$ > 0.05). However, there were significant gender differences in the frequency of ED consumption ($X^2$(1, N = 525) = 7.36, $p$ = .007) and mixing EDs with other substances ($X^2$(1, N = 413) = 10.65, $p$ = .001). Employment status showed no significant gender differences ($X^2$(3, N = 541) = 5.11, $p$ = .164), but work intensity had a significant relationship with gender ($X^2$(1, N = 541) = 8.01, $p$ = .005). The level of education did not differ significantly between males and females ($X^2$(4, N = 541) = 2.21, $p$ = .697). Alcohol consumption was significantly higher among males than females ($X^2$(1, N = 541) = 36.94, $p$ < .001), and there were no female smokers ($X^2$(1, N = 541) = 14.83, $p$ < .001) (Table 1).

### The prevalence of ED consumption

Fig 1 illustrates that 98.7% of youth in Tamale have ever consumed an ED, and 78.7% of them are current consumers (Fig 2).

**Table 1. Socio-demographic characteristics of sampled youth of Tamale.**

| Variable | Total (541) | Male (340) | Female (201) | P-value |
|---|---|---|---|---|
| **Age (yrs.), *n(%)*** | | | | .021 |
| 15–25 | 323 (59.70) | 213 (39.40) | 110 (20.30) | |
| 26–35 | 128 (23.70) | 82 (15.20) | 46 (8.50) | |
| 36–45 | 90 (16.60) | 45 (8.30) | 45 (8.30) | |
| **Age, *mean (SD)*** | 25.55 (8.11) | 24.98 (7.55) | 36.51 (8.92) | .034 |
| **Weight, *mean (SD)*** | 61.65 (9.08) | 63.69 (8.31) | 58.3532 (9.39) | < .001 |
| **Height, *mean (SD)*** | 1.64 (.10) | 1.66 (0.11) | 1.6019 (.07) | < .001 |
| **BMI, *mean (SD)*** | 23.19 (3.86) | 23.40 (3.85) | 22.8418 (3.86) | .105 |
| **Marital status, *n(%)*** | | | | .335 |
| Never Married | 382 (70.61) | 249 (73.20) | 133 (66.20) | |
| Married/Cohabiting | 141 (26.06) | 82 (24.10) | 59 (29.40) | |
| Separated | 8 (1.48) | 5 (1.50) | 3 (1.50) | |
| Divorced | 2 (.37) | 1 (.30) | 1 (.50) | |
| Widowed | 8 (1.48) | 3 (.90) | 5 (2.50) | |
| **Ethnicity, *n(%)*** | | | | .001 |
| Dagomba | 298 (55.10) | 211 (39.0) | 87 (16.10) | |
| Others | 243 (44.90) | 129 (23.8) | 114 (21.10) | |
| **Religion, *n(%)*** | | | | .059 |
| Christianity | 181 (33.50) | 100 (18.50) | 81 (15.00) | |
| Islam | 357 (66.00) | 238 (44.00) | 119 (22.00) | |
| Traditional | 3 (.50) | 2 (.40) | 1 (.10) | |
| **Employment status, *n(%)*** | | | | .164 |
| Student | 228 (42.10) | 134 (39.40) | 94 (46.80) | |
| Unemployed | 49 (9.10) | 28 (8.20) | 21 (10.40) | |
| Self-employed | 143 (26.40) | 99 (29.10) | 44 (21.90) | |
| Employed | 121 (22.40) | 79 (23.20) | 42 (20.90) | |
| **Work intensity, *n(%)*** | | | | .005 |
| Yes | 353 (65.20) | 237 (69.70) | 116 (57.50) | |
| No | 188 (34.80) | 103 (30.30) | 85 (42.30) | |
| **Education level, *n(%)*** | | | | .697 |
| None | 44 (8.10) | 29 (8.50) | 15 (7.50) | |
| Primary | 28 (5.20) | 19 (5.60) | 9 (4.50) | |
| JHS | 80 (14.80) | 49 (14.40) | 31 (15.40) | |
| Secondary/vocational | 184 (34.00) | 121 (35.60) | 63 (31.30) | |
| Tertiary | 205 (37.90) | 122 (35.90) | 83 (41.30) | |
| **Alcohol intake, *n(%)*** | | | | < .001 |
| Yes | 90 (16.60) | 82 (24.10) | 8 (4.00) | |
| No | 451 (83.40) | 258 (75.90) | 193 (96.00) | |
| **Smoking, *n(%)*** | | | | < .001 |
| Yes | 24 (4.40) | 24 (4.40) | 0 (.00) | |
| No | 517 (95.60) | 316 (92.90) | 201 (100.00) | |

## Consumption pattern of EDs

The analysis of the data revealed that the highest frequency of ED consumption among the study sample was weekly (70.9%), with smaller proportions consuming daily (36.9%) or monthly (2.2%) (Table 2). The majority of consumers (93.9%) consumed one bottle per sitting,

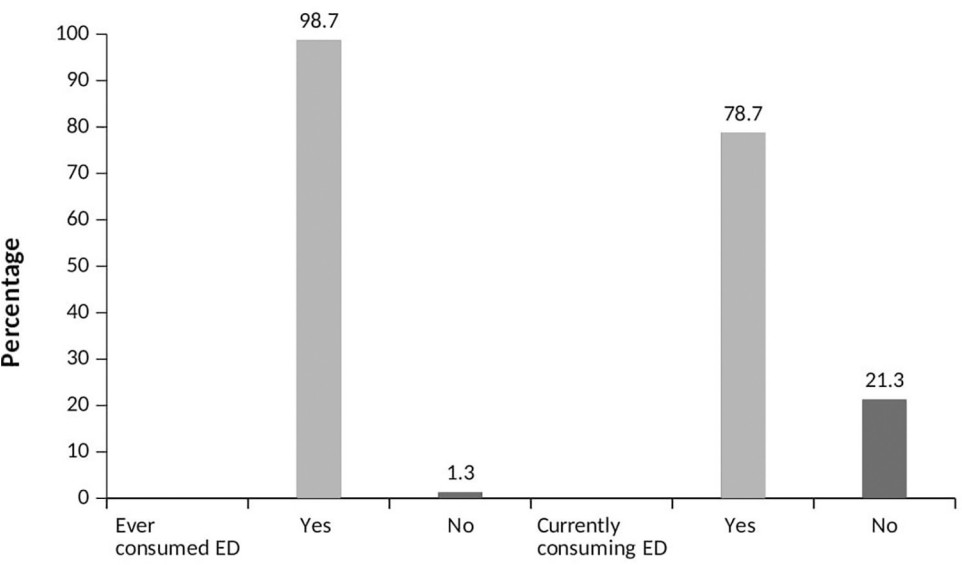

**Fig 2. Prevalence of ED consumption.**

Table 2. Consumption pattern among the youth in Tamale.

| Variable | Frequency (n) | Percentage (%) |
|---|---|---|
| **Frequency of ED consumption** | | |
| Daily | 111 | 26.90 |
| Weekly | 293 | 70.90 |
| Monthly | 9 | 2.20 |
| **Number of bottles drank per sitting** | | |
| One | 388 | 93.90 |
| Two | 17 | 4.10 |
| Three | 5 | 1.20 |
| Four | 3 | 0.70 |
| **Duration since consumption commenced** | | |
| One year and below | 46 | 11.10 |
| 2 to 3 years | 256 | 62.00 |
| 4 years and above | 11 | 26.90 |
| **Preferred time of consumption** | | |
| Morning | 31 | 7.50 |
| Afternoon | 155 | 37.50 |
| Evening/ night | 227 | 55.00 |
| **Reason for preferred time for consumption** | | |
| Energy boost for work or studies | 83 | 20.10 |
| To relax or refreshment | 32 | 7.70 |
| No reason | 119 | 28.80 |
| Manage fatigue and stress | 76 | 18.50 |
| Keep awake and concentrate | 56 | 13.60 |
| Craving | 27 | 6.50 |
| For its effects to dissipate by a certain time | 20 | 4.80 |

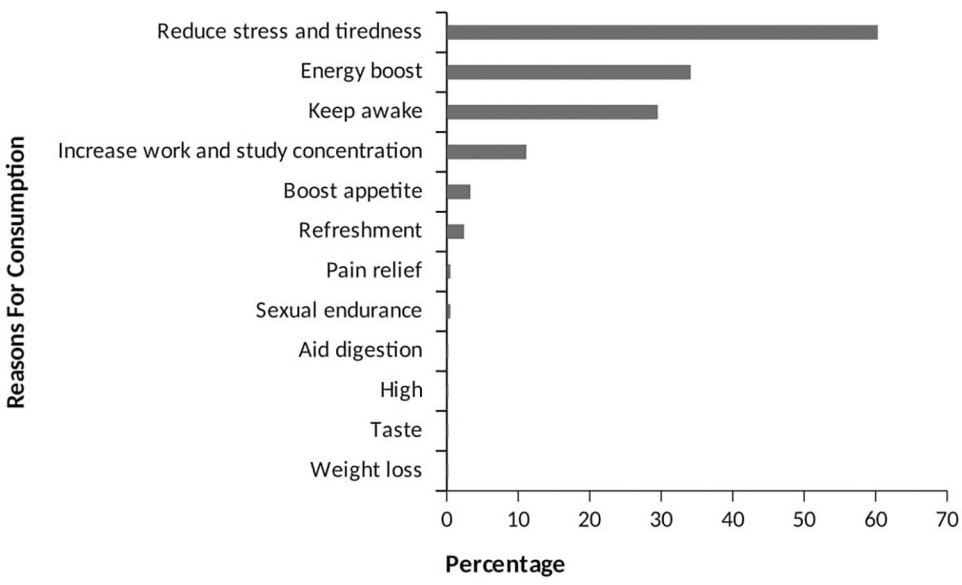

**Fig 3. Reasons for ED consumption.**

while smaller proportions consumed two (4.1%), three (1.2%), or four (0.7%) bottles/cans. The evening was the preferred time for consumption (55.0%), and most consumers (62.0%) had been consuming EDs for 2 to 3 years without a specific reason (Table 2). Fig 3 shows that the primary reasons for consuming EDs were to reduce stress and tiredness (60.3%), followed by seeking an energy boost (34.1%) and the desire to stay awake (29.5%). Smaller proportions mentioned other reasons such as increasing concentration, boosting appetite, refreshment, pain relief, sexual endurance, aiding digestion, seeking a high, enjoying the taste, and facilitating weight loss. Only 31.26% of respondents encountered EDs at social gatherings. The majority encountered them at weddings (50.3%), while 39.4% encountered them at outdoor events, and a smaller proportion encountered them at parties (Figs 4 and 5). Among the respondents, 25.9% experienced side effects. Insomnia was the most commonly reported side effect (55.1%), followed by post-consumption fatigue (18.1%), palpitations (16.8%), and restlessness (13.1%) (Figs 6 and 7).

## Knowledge and perception of ED consumption

Almost all of the respondents had heard about EDs (97.8%). The source of their information on EDs mainly came from Television (87.3%), friends (83.2%), the internet (51.6%), and radio (41.4%). With regards to known ingredients used for manufacturing EDs, Caffeine (76.4%), Taurine (24.6%), Guarana (24.4%), and sugar (38.4%) were considerably well known compared to others like ginseng (3.6%), and vitamins (5.1%). The most popular brand was Rush (93.2%) (Table 3). The most perceived benefit of taking EDs was to provide extra energy (81.0%). A greater proportion (81%) of the respondents perceived that the benefit of EDs is to get extra energy and about sixty-two per cent perceive stress reduction as a benefit. About sixty-one per cent of the sample perceived that the adverse effect of EDs is insomnia (Table 3). The highest proportion of respondents (83.4%) had poor knowledge of EDs. A comparatively smaller proportion had good and excellent knowledge of EDs (15.7% and .9% respectively) (Fig 8).

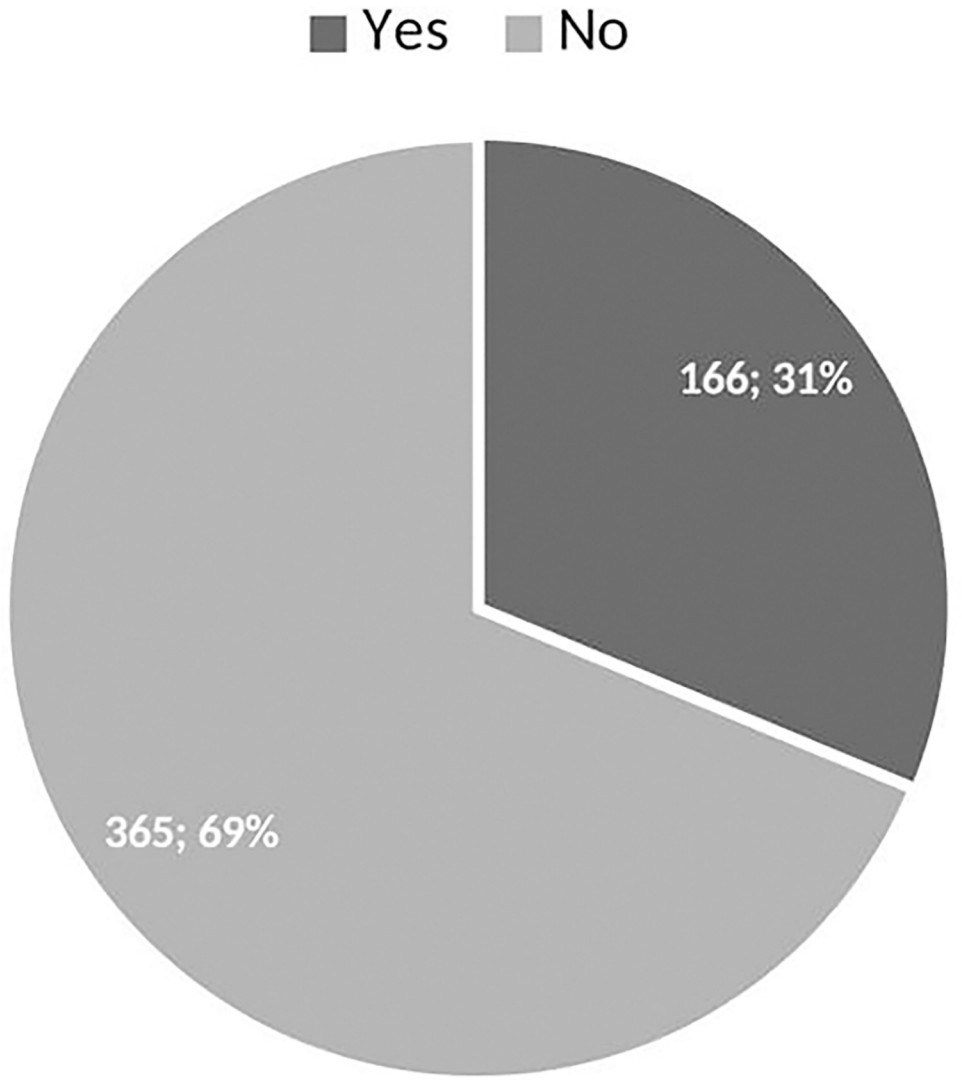

**Fig 4. The proportion of ED sightings at social gatherings.**

## Relationship between ED use and socio-demographic characteristics

An independent samples T-test revealed that consumers ($M = 24.4$, $SD = 7.6$) were generally younger than non-consumers ($M = 29.5$, $SD = 8.6$), $t(160.79) = -5.64$, $p < .001$. The Chi-square analysis indicated that the proportion of ED consumers decreased across different age categories: 264 (83.8%), 102 (82.9%), and 47 (54%) for the age groups 15–25, 26–35, and 36–45, respectively. Notably, each age group had a higher proportion of consumers compared to non-consumers, and the youngest age category (15–25) exhibited the highest proportion of users. These frequency differences were statistically significant, $X^2(2, N = 525) = 37.8$, $p < .001$. The Chi-square test indicated a significantly higher proportion of male consumers (66.6%) compared to female consumers (33.4%), $X^2(1, N = 525) = 7.36$, $p = .007$. Educational level was also found to be significantly associated with consumption, $X^2(4, N = 525) = 19.14$, $p = .001$. Additionally, respondents with high work intensity were more likely to consume EDs, $X^2(1, N = 525) = 24.03$, $p < .001$. Although there was no significant association between consumption and alcohol intake, $X^2(1, N = 525) = 1.299$, $p = .254$, a significant association was found

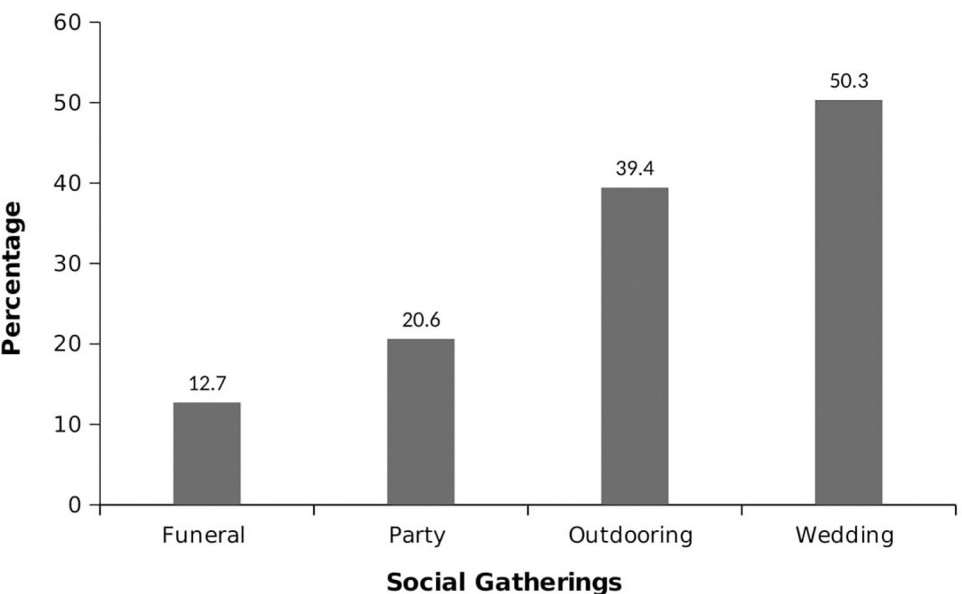

**Fig 5. Social gatherings where EDs have been sighted.**

between consumption and smoking, $X^2(1, N = 525) = 6.196$, $p = .013$. Non-smokers (96.6%) had a considerably higher proportion of consumers compared to smokers (3.4%) (Table 4).

## Risk factors related to ED consumption

A logistic regression analysis was conducted to examine the predictive value of various factors, including age, gender, marital status, education level, work intensity, alcohol intake, smoking, BMI, ED consumption at social gatherings, and knowledge of EDs, on the likelihood of consuming EDs. The logistic regression model yielded significant results, $X^2(23, N = 525) = 149.17$, $p < .001$. The model accounted for 24.7% (Cox & Snell R2) to 38.3% (Nagelkerke R2) of the variance in consumption and accurately classified 83.0% of cases. Table 5 demonstrates that marital status, education level, work intensity, servings at gatherings, and knowledge level of EDs significantly contributed to the model.

Regarding specific variables, individuals in the 26–35 age group had 0.359 times lower odds of not consuming EDs compared to those in the 15–25 age group ($p = .014$, 95% CI .159, .814). On the other hand, individuals in the 36–45 age group had 1.515 times higher odds of not consuming EDs than those in the 15–25 age group ($p = .398$, 95% CI .578, 3.975). Separated individuals had 12.626 times greater odds of not consuming EDs compared to singles ($p = .007$, 95% CI 2.001, 79.644). Furthermore, tertiary-educated individuals had 3.634 times higher odds of not consuming EDs than those without formal education ($p = .025$, 95% CI 1.175, 11.237). Individuals with low work intensity had 3.692 times higher odds of not consuming EDs compared to those with high work intensity ($p < .001$, 95% CI 2.055, 6.634). Moreover, the likelihood of consuming EDs was 84% higher when consumed privately compared to being served at public gatherings ($p < .001$, 95% CI .092, .277). Additionally, individuals with good knowledge of EDs were 2.714 times more likely to not consume EDs than those with poor knowledge ($p = .004$, 95% CI 1.375, 5.348). However, the 36–45 age category, gender, marital status (married, divorced, widowed), work type, alcohol intake, smoking, BMI, and excellent knowledge of EDs did not have a significant impact on the model (Table 5).

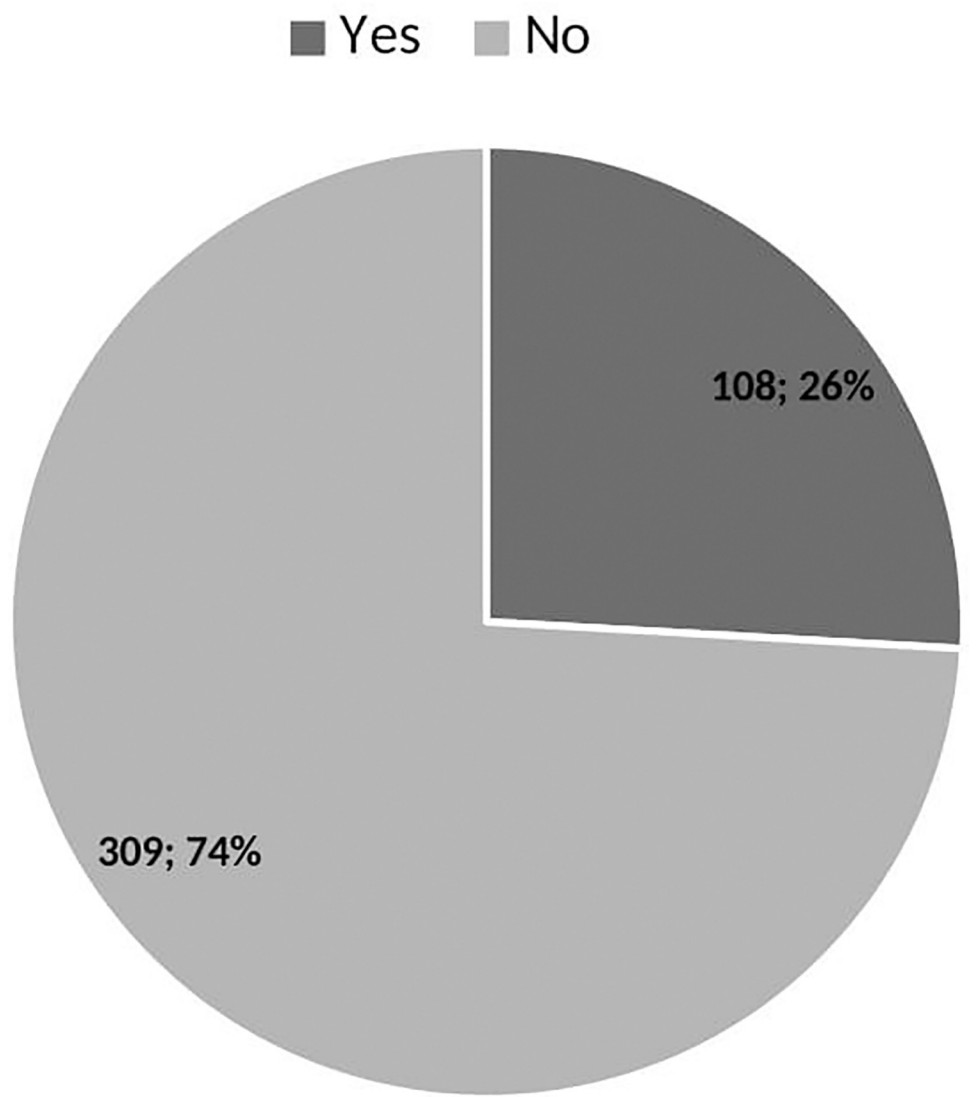

**Fig 6. The proportion of ED users who have experienced side effects.**

## Discussion

EDs are gaining popularity, but studies have linked them to harmful effects. Despite this, more people are consuming them for various reasons. Given the lack of policies in Ghana and the aggressive marketing by ED companies, it is crucial to understand consumption patterns and predictors. This knowledge can help inform policies and educational efforts by public health professionals to address this growing phenomenon. Hence, this study aims to evaluate the youth's perceptions, consumption habits, and factors linked with ED consumption in the Tamale Metropolis.

### Knowledge

According to this research, a high percentage of the study population is aware of EDs, consistent with other studies [19, 26]. The perceived benefits of these drinks align with their advertised effects, suggesting strong marketing efforts. However, negative effects are often learned

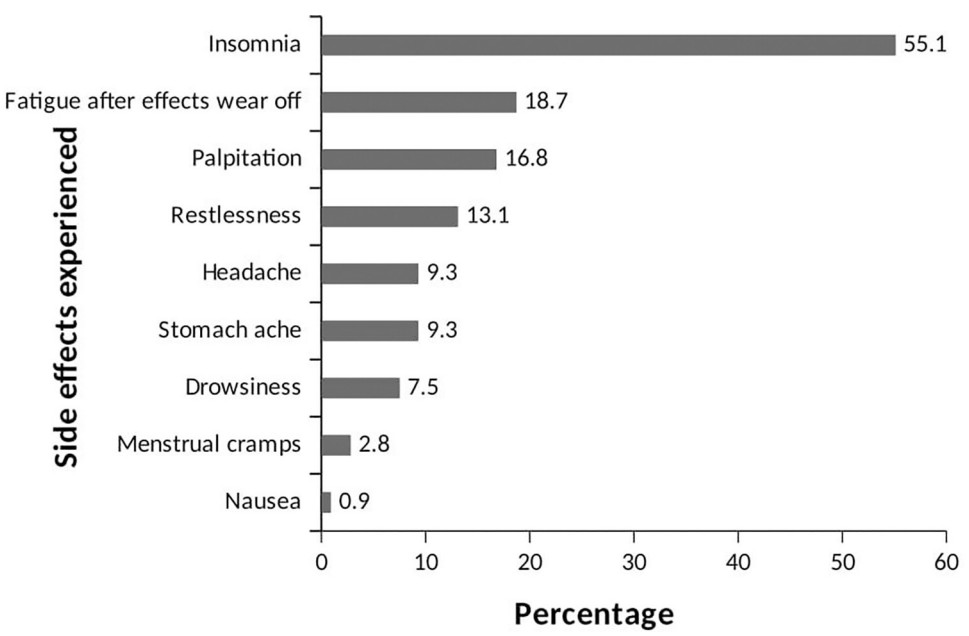

**Fig 7. Side effects experienced by ED users.**

through personal experiences rather than advertising. Therefore, obtaining information from friends and relatives can have a more positive impact on decisions about ED consumption compared to direct advertising. Other studies have also shown the influence of friends, family, and advertisements on awareness of EDs, but the nature of this influence (positive or negative) remains undetermined [19, 27]. Increased brand awareness is a result of the multifaceted media channels employed for the promotion of EDs. Producers of EDs commit substantial resources to advertising and promotional strategies encompassing television broadcasts, public signage, posters, and sponsorship initiatives [6, 22, 28, 29]. Also, the significance of social factors, like friends, family, and advertising, in shaping both awareness and consumption patterns cannot be overstated. Within the framework of the Health Belief Model (HBM) [30], this underscores the pivotal role of social influences, acknowledging their impact on interpersonal relationships and, consequently, health-related decision-making processes.

This study found that the popularity of EDs is highly driven by advertising media, especially television. In support of this assertion, Stacey et al. [31] substantiated a correlation between the consumption of EDs and the exposure to promotional advertisements of ED brands. These advertisements were found to typically target a youthful male audience using sports and entertainment-centered themes [32], and leveraging associations with influential celebrities. Gendered advertising reflects the leveraging of higher involvement of males in sports. This is in line with an American study that strongly links ED consumption to traditional masculinity and risk-taking tendencies [33].

According to a study by Coso et al. [34], the most common ingredients found in EDs, comprising over 50% of the content, are caffeine, vitamin B6, sodium, niacin, and vitamin B12. Caffeine is the most well-known ingredient, with 76.4% awareness among consumers. However, there is a contrast in awareness of other ingredients such as taurine (24.6%), sugar (38.4%), and guarana when compared to similar studies [26]. The popularity of caffeine awareness can be attributed to its prominent marketing as a key ingredient in EDs. Lesser-known ingredients are not heavily marketed and may be too technical for consumers to be concerned

**Table 3. Knowledge and perception of the sample on ED consumption.**

| Variable | Frequency (n) | Percentage (%) |
|---|---|---|
| **Ever Heard of Eds** | | |
| Yes | 529 | 97.80 |
| No | 12 | 2.20 |
| **Heard About EDs Through** | | |
| Television | 462 | 87.30 |
| Radio | 219 | 41.40 |
| Internet/social media | 273 | 51.60 |
| Friends | 440 | 83.20 |
| Billboard | 11 | 2.10 |
| **Known ED Ingredients** | | |
| Guarana | 129 | 24.40 |
| Taurine | 130 | 24.60 |
| Salt | 24 | 4.50 |
| Caffeine | 404 | 76.40 |
| Water | 326 | 61.60 |
| Sugar | 203 | 38.40 |
| Ginseng | 19 | 3.60 |
| Color | 6 | 1.10 |
| Flavor | 20 | 3.80 |
| Vitamins | 27 | 5.10 |
| Alcohol | 45 | 8.50 |
| Acid | 19 | 3.50 |
| **Known ED Brands** | | |
| Storm | 468 | 88.10 |
| 5-Star | 472 | 88.90 |
| Rush | 495 | 93.20 |
| Burn | 37 | 6.80 |
| Red Bull | 46 | 8.70 |
| Blue Jeans | 119 | 22.40 |
| Tamarinda | 1 | .20 |
| Power Up | 5 | .90 |
| Explode | 10 | 1.90 |
| Rox | 53 | 10.00 |
| Passion | 12 | 2.30 |
| Easy Sport | 123 | 23.20 |
| Run | 146 | 27.30 |
| Lucozade | 31 | 5.80 |
| Vim | 56 | 10.40 |
| 10–10 | 66 | 12.20 |
| Boss | 16 | 3.00 |
| Vody | 26 | 4.90 |
| Bullet | 17 | 3.20 |
| Next Level | 12 | 2.30 |
| **Perceived ED Benefits** | | |
| Increase work/study concentration | 106 | 20.00 |
| Give extra energy | 430 | 81.00 |
| Boost Appetite | 61 | 11.50 |

(*Continued*)

**Table 3.** (Continued)

| Variable | Frequency (n) | Percentage (%) |
|---|---|---|
| Reduce stress | 331 | 62.30 |
| Nutrient source | 1 | .20 |
| Prevents malaria | 1 | .20 |
| Reduce or manage hunger | 1 | .20 |
| High | 1 | .20 |
| Pain killer | 5 | .90 |
| Sexual stamina | 5 | .90 |
| Aid digestion | 2 | .40 |
| Increase work rate | 1 | .20 |
| Lose weight | 2 | .40 |
| Keep awake | 48 | 9.00 |
| Refreshment | 4 | .80 |
| **Perceived Side Effects of Eds** | | |
| Insomnia | 322 | 60.60 |
| Drowsiness | 59 | 11.10 |
| Restlessness | 273 | 51.40 |
| Headache | 105 | 19.80 |
| Addiction | 20 | 3.80 |

about. This indicates that consumers of processed products, including EDs, prioritize the perceived or real benefits of the product rather than paying attention to its specific contents. Individuals may not fully understand the information provided on ED labels or simply place a high level of trust in the producers.

Rush, 5-Star, and Storm are the most popular ED brands in the Tamale Metropolitan area, with high levels of popularity ranging from 88.1% to 93.2%. In contrast to this study, prior

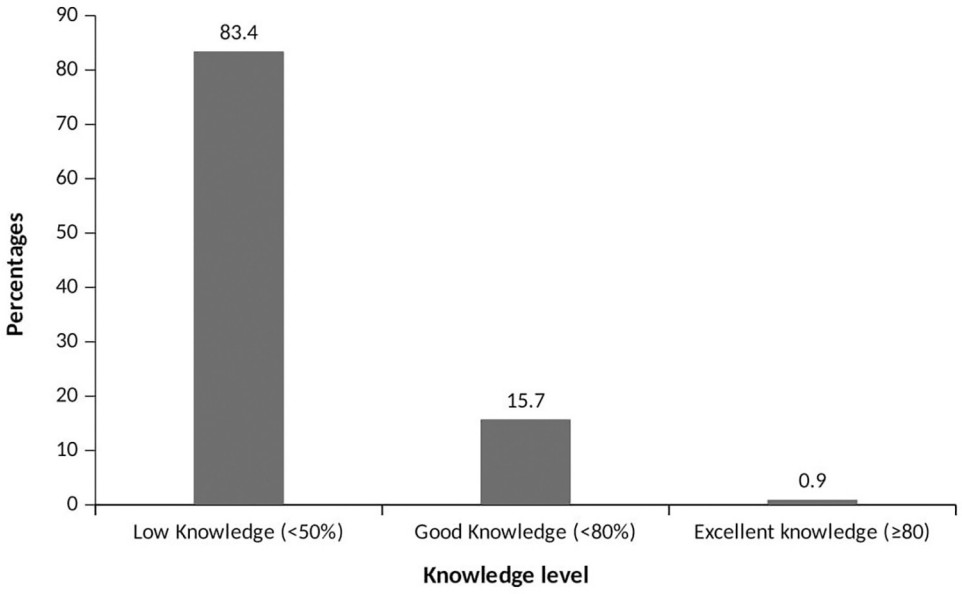

**Fig 8. Knowledge level of respondents.**

**Table 4. Association between ED use and socio-demography.**

| Variables | ED consumption, Yes (N = 413) | ED consumption, No (N = 112) | P-value |
|---|---|---|---|
| **Age, *mean (SD)*** | 24.41 (7.62) | 29.47 (8.64) | < .001 |
| **Age Categorized, *n (%)*** | | | < .001 |
| 15–25 | 264 (63.90) | 51 (45.50) | |
| 26–35 | 102 (24.70) | 21 (18.80) | |
| 36–45 | 47 (11.40) | 40 (35.70) | |
| **Gender (M/F), *n (%)*** | | | .007 |
| Male | 275 (66.60) | 59 (52.70) | |
| Female | 138 (33.40) | 53 (57.30) | |
| **Educational level, *n (%)*** | | | .001 |
| None | 36 (8.70) | 8 (7.10) | |
| Primary | 25 (6.10) | 2 (1.80) | |
| JHS | 66 (16.00) | 12 (10.70) | |
| Secondary/vocational | 151 (36.60) | 29 (25.90) | |
| Tertiary | 135 (32.70) | 61 (45.50) | |
| **Work type, *n (%)*** | | | .066 |
| Employed | 85 (20.60) | 33 (29.50) | |
| Self-employed | 111 (26.90) | 28 (25.00) | |
| Unemployed | 32 (7.70) | 13 (11.60) | |
| Student | 185 (44.80) | 38 (33.90) | |
| **Work intensity, *n (%)*** | | | < .001 |
| Yes | 294 (71.20) | 42 (46.40) | |
| No | 119 (28.80) | 60 (53.60) | |
| **Alcohol intake, *n (%)*** | | | .254 |
| Yes | 66 (16.00) | 23 (20.50) | |
| No | 347 (84.00) | 89 (79.50) | |
| **Smokers, *n (%)*** | | | .013 |
| Yes | 14 (3.40) | 10 (8.90) | |
| No | 399 (96.60) | 102 (91.10) | |

research conducted in Ghana has yielded dissimilar results, with the most frequently consumed ED brands being identified as Blue Jeans, Lucozade, Rox, Monster, and Red Bull [20, 21]. The variation in popularity could be influenced by factors such as economic and demographic conditions, product effectiveness, taste, and availability [32, 35]. Therefore, marketing efforts, affordability, and favourable taste of Rush, 5-Star, and Storm brands likely contribute to their popularity in the Tamale Metropolitan area, especially among the youth who prefer these drinks over more expensive premium brands like Red Bull and Blue Jeans.

EDs are perceived to have several benefits, including increased alertness, improved work or study concentration, energy boost, stress management, and promotion of health and social relationships [15]. This study aligns with previous research [22, 27, 36, 37], indicating a prevailing belief in the benefits of EDs, with the highest perceived advantages being increased work or study concentration (20%), energy boost (81%), and stress management (62.3%). Also, this study reported perceived adverse effects including insomnia (60.60%), and restlessness (51.40%) as commonly reported adverse effects of EDs, similar to work by Casuccio et al. [26]. While perception is known to be autonomous concerning thought, knowledge may lead to an intentional intervention in the process of achieving a percept [38]. This suggests that perceptions may be influenced by external factors, notably well-targeted advertisements and recommendations from friends and family. Understanding these external influences is crucial for

**Table 5. Logistic regression predicting the likelihood of ED consumption.**

| Variables | | OR | 95% C.I. | | P-value |
|---|---|---|---|---|---|
| | | | Lower | Upper | |
| **Age in categories** | 15–25 | 1 | | | .002 |
| | 26–35 | .359 | .159 | .814 | .014 |
| | 36–45 | 1.515 | .578 | 3.975 | .398 |
| **Gender** | Male | 1 | | | |
| | Female | 1.652 | .951 | 2.869 | .075 |
| **Marital status** | Single | 1 | | | .043 |
| | Married | .872 | .392 | 1.942 | .738 |
| | Separated | 12.626 | 2.001 | 79.644 | .007 |
| | Divorced | .925 | .003 | 319.780 | .979 |
| | Widowed | 3.208 | .354 | 29.027 | .300 |
| **Education level** | None | 1 | | | .001 |
| | Primary | .465 | .075 | 2.890 | .411 |
| | JHS | 1.069 | .287 | 3.979 | .921 |
| | SHS/ vocational | .993 | .316 | 3.124 | .991 |
| | Tertiary | 3.634 | 1.175 | 11.237 | .025 |
| **Type of work** | Employed | 1 | | | .220 |
| | Self-employed | 1.624 | .747 | 3.532 | .221 |
| | Unemployed | 1.130 | .398 | 3.208 | .818 |
| | Student | .700 | .329 | 1.490 | .355 |
| **Work intensity** | Yes | 1 | | | |
| | No | 3.692 | 2.055 | 6.634 | < .001 |
| **Alcohol intake** | Yes | 1 | | | |
| | No | .835 | .387 | 1.802 | .646 |
| **Smoking** | Yes | 1 | | | |
| | No | .672 | .192 | 2.359 | .535 |
| **BMI categories** | Underweight | 1 | | | .941 |
| | Normal | .713 | .242 | 2.098 | .539 |
| | Overweight | .696 | .206 | 2.356 | .560 |
| | Obese | .730 | .183 | 2.910 | .655 |
| **EDs served at gatherings** | Yes | 1 | | | |
| | No | .160 | .092 | .277 | < .001 |
| **Knowledge level** | Low knowledge | 1 | | | .010 |
| | Good knowledge | 2.714 | 1.378 | 5.348 | .004 |
| | Excellent knowledge | 4.295 | .467 | 39.493 | .198 |

devising effective health communication strategies and interventions to promote informed and balanced perspectives on ED consumption.

Further, upon analyzing the data, it is evident that a large proportion of the sample population in this study has poor knowledge of EDs, accounting for 83.4%. This trend is consistent with another study that also reported a high rate of poor knowledge among participants [19]. The implication of low knowledge is that it can result in a high prevalence of ED use. Therefore, it is crucial to address issues of misconceptions and lack of knowledge. For instance, information about the adverse effects of EDs can be incorporated into default marketing strategies. Also, the lack of warning labels and inadequate inclusion of risks and side effects in advertisements and messaging contribute to this poor knowledge [19, 32, 39].

## Attitude and practice

In Denmark and other places, consumption prevalence among young adults is low (15.8%) due to these products being relatively new and more expensive, compared to other soft drinks [33, 40]. Conversely consumption of EDs is high (78.7%) among the youth in Tamale. This finding aligns with studies conducted in Sicily among adults (78%), Ho, Ghana among commercial drivers (75%) and student-athletes in Ghana (62%) [19, 20, 26]. Moreover, an even higher percentage of individuals have ever tried EDs, with nearly the entire study population (98.7%) reporting consumption at some point. This trend is also consistent with studies conducted among adolescents in Shanghai, China (70.5%) and in Ho (85.6%) [19, 41]. The high prevalence of ever-consuming EDs is attributed to factors such as curiosity and perceived utility [24, 36]. Their increasing popularity is also attributed to various factors, including individual factors such as perceived benefits, targeted marketing strategies, and limited knowledge about these beverages [21, 32, 42]. Additionally, interpersonal factors like peer pressure, social image, and parental influence also play a role, along with environmental factors such as affordability, availability, and the lack of policies and regulations [41]. In the context of the health belief model (HBM) [30], these external factors serve as triggers of ED consumption. They can significantly impact individuals' decision to consume EDs. Therefore, such channels of influence must be monitored to ensure that the general public is well-informed to make healthy choices. ED consumption among Ghanaian youth is concerning due to its potential health risks [43]. These drinks have high levels of caffeine and sugar, which can cause cardiovascular and neurological issues and contribute to weight gain, obesity, and dental problems [16, 44–48]. Therefore, there is the need to place more emphasis on the need to address the health implications of ED consumption, particularly for the youth.

The main reasons for consuming EDs reported by the respondents were performance enhancement, stress management, energy boost, and increased work or study concentration. In similar studies conducted among drivers, students and athletes, it was reported that EDs were used to manage fatigue, stay awake, and concentrate for extended periods [19, 49–51]. Many ED users appreciate their ability to reduce sleep hours, enhance concentration, boost energy, and serve as refreshments. Particularly, workers use EDs to enhance performance in the workplace, while students use them to cope with academic pressure [19, 50, 51]. Also, withdrawal symptoms have been reported by ED consumers after the effects wore off. Such an outcome potentially leads to dependence and workplace accidents–a risk to users and others [52, 53]. However, non-users tend to avoid them due to awareness of potential side effects [22]. This aligns with the HBM [30] consideration of perceived benefits for users and an understanding of perceived susceptibility and severity for non-users. A sub-group of ED users in this study reported mixing water with EDs to reduce their effects. This is a positive health behavior that demonstrates responsible consumption and reflects the perceived control over their actions. Adverse effects experienced by consumers included abdominal pain, chest pain, fatigue, constipation, insomnia, palpitations, diuresis, and muscle weakness. Although a wide range of adverse reactions have been reported by ED users, a small proportion experience side effects according to this study. This is possibly due to responsible consumption patterns within the sample population. Compared to the perceived benefits highlighted by the HBM [30], the adverse effects and the influence of personal experiences, also reveal potential barriers to ED consumption.

EDs are particularly popular among young individuals, especially those in their early twenties [32, 36, 54]. This can be attributed to the energy-demanding nature of that age group [55]. Young people's curiosity [56] and inclination to follow trends also contribute to their higher consumption of EDs compared to older individuals [43]. Additionally, males generally

consume EDs more frequently, particularly at night, and experience more side effects. This aligns with the marketing strategies of EDs, which often target young males and emphasize activities associated with youth and masculinity, such as sports [6, 31, 36, 54, 57, 58].

Education level is associated with ED use [26, 59]. Accordingly, ED consumption is higher among those with low education levels, and people who work or study long hours [19]. Education has an effect on lifestyle choices and awareness which reflects in consumption patterns. Individuals with a higher work intensity are more likely to consume EDs as a coping strategy against increasing demands. This supports the notion that activities like work and academics influence calorie intake and, consequently, ED consumption [26, 59, 60]. Intense work is typically characterized by physically demanding tasks, long work hours, and limited breaks. Individuals engaged in intense work or educational activities find EDs to be a convenient way to boost energy without interrupting their work for a meal. Moreover, while EDs may impact energy levels, they may not provide sufficient nutritional value and satiety. Their consumption may also lead to decreased reliance on natural foods and inadequate rest management [61]. Variables such as age, gender, education level, and work intensity, are factors that influence patterns of ED consumption. This aligns with the HBM [30], where the acknowledgement of modifying variables is pivotal in comprehending their role in shaping health-related behaviors.

According to this study, smokers make up a small proportion of ED consumers. The lack of association between alcohol consumption and ED use is attributed to taste preferences and concerns about the impact of combining the two for performance [62]–one of the primary reasons for consuming EDs among the participants in this study. Similar to previous studies by Casuccio et al. [26] and Chang et al. [15], no significant associations were found between ED use and these variables. Furthermore, this study revealed that easy access to EDs corresponds to higher consumption rates. EDs are readily available and easily accessible [63] to children and young adults, with numerous brands found in every shop that sells beverages [47, 64]. Additionally, there is a lack of government regulations governing the sale and marketing of EDs. This unrestricted promotion and utilization of EDs contributes to their increasing appeal, popularity, and availability [5]. The knowledge, experience of side effects or benefits also plays a role in influencing the use of EDs [64] Knowledge about EDs tends to increase with age [65]. And this reflects in the likelihood of ED consumption with increasing age [26, 66, 67]. With age individuals become more conscious of their health. This is because they generally do not believe in the durability of their bodies compared to younger people.

Among the participants in this study, a high proportion (70.9%) consumed EDs weekly. A smaller percentage consumed them daily (26.9%) or monthly (2.2%). In contrast, a study by Subaiea et al. [32] found a higher proportion of monthly consumers and fewer weekly and daily consumers. However, there are differences in the age range and age distribution between the two studies, which may explain the variation in consumption frequency patterns. Other studies have reported that most consumers tend to consume EDs weekly and have one drink per day [49, 58, 68].

Interestingly, the majority of consumers in this study started consuming EDs within the past 2 to 3 years. This coincides with the introduction of new and affordable ED brands in the Ghanaian market, suggesting that these factors contribute to the increasing popularity of EDs. The frequency of consumption is also influenced by the relatively low price of EDs, which makes them easily accessible to individuals with lower purchasing power [5, 69].

While another study suggests that ED consumers do not have a preferred time of consumption, the participants in this study showed a tendency to consume EDs in the evenings or at night [32]. This finding aligns with the study conducted by Chang et al. [15], where more than half of the consumers reported consuming EDs at night. This preference for evening consumption may be related to specific usage patterns, particularly among those who utilize them for studies, staying awake, or managing stress and fatigue. Consumers are less likely to consume

EDs in the morning when they are already rested and do not require an energy boost. However, consumption rates tend to increase in the afternoon when workers start to feel tired from their morning activities [19].

ED consumption has been associated with smoking, alcohol intake, cannabis, medical drugs, and opioid use in previous studies [59, 70]. However, EDs pose risks to mental health, cardiovascular health, dental health, metabolic health, and social well-being [59, 70–72]. While another study has reported an association between ED use and alcohol intake [62, 73], this study's results align with a study by Peacock et al. [54]. It found that there is no significant association between ED use, smoking and alcohol consumption. However, this study's population consists of few alcohol users and smokers likely due to the strong religious affiliations among the participants. This reflects on the habit of mixing EDs with other substances. The majority of users and non-users of EDs do not support the idea of mixing them with other substances, as supported by the study conducted by Chang et al. [15]. When it comes to mixing EDs with other substances, consumption at social gatherings is implicated in the promotion of such behavior compared to consumption at home [74]. While this study agrees with Casuccio et al. [26] that public places and events may not greatly influence ED consumption, other studies indicate that attention must be invested in those influencers [2, 75].

According to the findings of this study, individuals experience various side effects when the effects of EDs wear off. These side effects include insomnia, fatigue, palpitations, restlessness, headaches, stomach aches, drowsiness, menstrual cramps, and nausea. It is worth noting that these results differ slightly from a previous study conducted by Subaiea et al. [32], which examined the consumption patterns of EDs among the Saudi population. The variation in reported side effects may be attributed to the different brands of EDs consumed by each population.

## Predictors of ED consumption

The findings of this study show certain factors may influence ED consumption. Consistent with the findings of other studies [76–78], age is a significant predictor of ED consumption. For those in the younger age groups, consumption can be associated with developmental influences on consumption behaviors such as curiosity, addiction, peer influence, high perception of their body's durability, or low health consciousness. In the older age group, consumption indicates shifts in preferences or lifestyle factors as individuals enter adulthood, greater purchasing power, increased dependence on external sources of energy, and increased energy demands. However, generally, ED consumption decreases with increasing age [40, 79, 80]. Marital status is a predictor of ED consumption as well [81]. This appears to be the case because relationships with people play a key role in aspects of lifestyle and decision-making. By extension, it impacts nutrition and health-seeking behavior. Individuals with different marital statuses might have distinct motivations for consuming EDs, influenced by factors such as social norms, stress levels, marital conflicts, the stress of a failed marriage and other marriage dynamics. Therefore, it is not out of place that the findings of this study have revealed marital status to be a factor that influences consumption. The intensity or perceived intensity of work done by an individual inadvertently causes the need for more energy to complete tasks, especially for those in the blue-collar job bracket. This is in line with this study's findings that indicate that people who perceive their work to be low in intensity are less likely to consume EDs. Meanwhile, another study conducted among adolescents contradicts this finding by reporting that ED consumption is highest among those who are less physically active [40]. The difference may be a result of a difference in culture and the way adolescents are raised as well as the span of age categories covered in both studies. Education plays a key role in exposure and understanding of information including the types related to health. And this is demonstrated in the

findings—the level of education has a relationship with ED consumption. Those with tertiary education are less likely to consume EDs when compared to those who have never had a formal education [82]. This could mean that the exposure to and ability to seek and understand health information such as what EDs are and what their potential side effects may be, people with formal education are better informed to make the right choices. Those who have not had a formal education are usually unable to verify information, struggle to accept new information, are easily misinformed, accepting of myths, and discard facts in some cases. A related factor of interest is the level of knowledge people have of EDs in general. An understanding of the drinks referred to as EDs, their potential benefits, side effects, and even constituent ingredients could influence the decision to consume ED products. Meanwhile, a great proportion of individuals have low knowledge of EDs [82]. Special attention should be given to those with low education status and low health literacy because formal and health education can play a critical role in the way in which people consume EDs. Evidence from other studies shows that higher education is associated with reduced consumption of EDs [58, 82]. Consumption of EDs was found to be 84% higher when consumed in private compared to consumption at public gatherings. This may be the case because EDs are served with other drink options. They may be served per person according to budgeting constraints which will impact consumption. Also, on the other hand, it is the private intrinsic needs of individuals that may push them to consume more EDs.

## Recommendations for ED sale and consumption

In 2023, Ghana passed a tax bill on sugar sweetened-beverages into law. The law categorizes EDs as sugar sweetened drinks. The policy action is expected to increase individual savings and promote health [83]. Taxation is an effective strategy for reducing the consumption of sugar sweetened-beverages. While higher taxes have been more effective (above 10%), such initiatives have no significant impact in high-income countries compared to LMICs [84]. Therefore its application in Ghana is likely to yield positive results. The extant tax regulations exhibit commendable strides in mitigating the consumption of EDs; however, there exists untapped potential for further reduction. It is imperative to reconsider the categorization of EDs given that they differ from conventional sugar-sweetened beverages, as they have cardiovascular effects substantiated by several of compelling case studies [11, 85–89]. This reclassification stands to facilitate a spectrum of viable interventions, encompassing but not confined to the following: stringent restrictions on the pervasive retail distribution of EDs; the implementation of rigorous age-related restrictions, and prohibition in schools governing the purchase and consumption of EDs; enhanced oversight of marketing, advertising, and product labelling; and the unequivocal disclosure of potential short- and long-term adverse effects associated with the consumption of EDs [6].

In alignment with the national food-based dietary guidelines, which stipulate that individual sugar consumption should not surpass 50 grams per day, there is a pressing need for heightened advocacy efforts aimed at fostering decreased sugar intake and curbing the consumption of sugar-sweetened beverages [90]. As EDs are prohibited in school environments, natural fruit drinks should be encouraged. This can be incorporated into the national school feeding program initiative. Additionally, it is essential to provide backing and incentives to beverage manufacturers to invest in the production of healthy, nutrient-rich beverages characterized by low sugar content and essential nutrients.

## Limitations of the study

The adoption of a cross-sectional design restricts the establishment of causal relationships and temporal changes in ED consumption patterns and perceptions. The use of a convenience

sample raises concerns about representativeness, potentially limiting the generalizability of findings. Self-report data may introduce recall bias and may not fully reflect actual behaviors and beliefs accurately. The choice of the study area based on personal familiarity introduces potential bias in the selection process. Face-to-face interviews might also introduce interviewer bias and compromise participant anonymity.

## Conclusions

The study examined the prevalence and consumption of EDs among the youth in the Tamale Metropolis, along with their perceptions and factors influencing consumption. The findings indicate that more than half of the respondents currently consume EDs. However, there is a lack of knowledge regarding the ingredients, potential benefits, and harmful effects of EDs. Several factors were identified as independently associated with ED consumption, including age, marital status, work intensity, education level, knowledge of EDs, and the availability of EDs at public gatherings. This research highlights a high consumption and low abuse of EDs. This is despite a limited understanding of them, raising concerns about regulations related to advertising, marketing, distribution, and usage, as well as the need to quantify the size of the ED market. Unlike previous studies that focused on specific groups such as commercial drivers, students, and athletes, this study provides insights into ED use and knowledge a wider demographic.

## Author Contributions

**Conceptualization:** Williams Kobik.

**Data curation:** Williams Kobik.

**Formal analysis:** Williams Kobik.

**Investigation:** Williams Kobik.

**Methodology:** Williams Kobik.

**Project administration:** Williams Kobik.

**Resources:** Williams Kobik.

**Software:** Williams Kobik.

**Supervision:** Paul Armah Aryee.

**Validation:** Paul Armah Aryee.

**Visualization:** Williams Kobik.

**Writing – original draft:** Williams Kobik.

**Writing – review & editing:** Williams Kobik, Paul Armah Aryee.

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
