## [Decision Letter · Decision Letter 0]

19 Oct 2023

PONE-D-23-21713Title - The youth of tamale metropolis: understanding energy drink consumption, perceptions and related factorsPLOS ONE

Dear Dr. Kobik,

Thank you for submitting your manuscript to PLOS ONE. After careful consideration, we feel that it has merit but does not fully meet PLOS ONE’s publication criteria as it currently stands. Therefore, we invite you to submit a revised version of the manuscript that addresses the points raised during the review process.

We look forward to receiving your revised manuscript.

Kind regards,

Zakari Ali, PhD.

Academic Editor

PLOS ONE

Journal Requirements:

2. In the ethics statement in the Methods, you have specified that verbal consent was obtained. Please provide additional details regarding how this consent was documented and witnessed, and state whether this was approved by the IRB

3. "Thank you for stating the following financial disclosure: 

Additional Editor Comments:

Dear authors, we apologise for the delays in getting you feedback on your paper. We have now secured sufficient peer-review comments on your paper and are happy for you to revise based on their recommendations and submit for further evaluation. This paper is timely given the recent discussions from health and nutrition professionals in Ghana around soft and energy drinks consumption particularly among the youth.

However, it is clear from the two reviewers and from my own assessment as Editor that, your paper needs substantial proofreading to tighten up the arguments and correct language errors. Also make sure you back all claims with sufficient evidence.

1) In addition to comments from the reviewers, consider reducing sections of the discussion and introduce a new section in the discussion on policy measures/recommendations for EDs in Ghana. After reading your findings, a good question is, what next? And so what so we do, in terms of policy or action? Can you link this discussion to the recently released Ghana Food Based Dietary Guidelines? What should be done around schools? What (healthy) alternatives are there and why do you think those will be chosen over energy drinks?

2) What are the limitations of your study? Include this in the discussion. Reviewer 2 highlights some of these in different comments.

Reviewers' comments:

Reviewer's Responses to Questions

**Comments to the Author**

1. Is the manuscript technically sound, and do the data support the conclusions?

Reviewer #1: Yes

Reviewer #2: Yes

2. Has the statistical analysis been performed appropriately and rigorously? 

Reviewer #1: Yes

Reviewer #2: Yes

3. Have the authors made all data underlying the findings in their manuscript fully available?

Reviewer #1: Yes

Reviewer #2: Yes

4. Is the manuscript presented in an intelligible fashion and written in standard English?

Reviewer #1: Yes

Reviewer #2: Yes

5. Review Comments to the Author

Reviewer #1: This is indeed a very good and interesting Public health topic worthy of discussion. There are some suggestions which when added, can boost the strength of this research. They are as follows ;

Introduction

1.What previous works have been done in Ghana regarding investigation of energy drink consumption

2.What gaps were identified ?

3.How unique is this work or how different is this work from the previous works

4.A map of Ghana showing Tamale and coordinates must be provided

5.Last paragraph of the intro must be reconstructed for clarity

“This study will make…..

6.Discussion seems summary of the whole report. Authors need to discuss key findings with sufficient literature

7.Make the work aligned to the health belief model

Results

8.The paragraph before socio-demographic must be sent to the Methodology portion.

Reviewer #2: That you for the opportunity to review your work. The article provides valuable insights into the consumption patterns and perceptions of energy drinks among the youth in the Tamale Metropolis. I find the work valuable and pertinent to the research landscape around energy drinks however, will benefit from some minor revisions. Find below my general and specific comments to improve the quality of your work.

General comments

Ensure consistency in terms used throughout the paper. For example, "energy drinks" and "EDs" are used interchangeably. While abbreviations are acceptable, introduce them properly the first time and use them consistently thereafter.

Ensure that all claims, especially those that are not common knowledge, are backed by appropriate references.

Maintain consistency in terms, abbreviations, and formatting throughout the paper.

Some sentences are lengthy and could be split for better readability and understanding.

The methods section, in particular, could benefit from more detailed explanations to demonstrate the rigor of the research process.

Ensure that all claims, especially statistical ones, are properly cited. For instance, statements like "62% of student-athletes consume at least one can of energy drink per week" should have a reference.

Specific Comments:

Abstract:

The sentence "Energy drinks were consumed by the majority of the youth" is redundant given the statistics provided earlier in the abstract. Consider removing or rephrasing.

Introduction:

The statement "Ingredients commonly found in energy drinks include caffeine, guarana, ginseng, taurine, glucose, L-carnitine, glucuronolactone, and B vitamins" could benefit from a citation.

The transition between the global context of energy drinks and the specific situation in Ghana could be smoother. Consider adding a sentence to bridge the two sections.

"Throughout history, humans have sought physical and cognitive enhancements to tackle challenging tasks."

This is a broad statement. Consider adding a citation or example for context.

"The rise of energy drinks (EDs) and their marketing strategies, aimed at boosting physical and cognitive performance, has led to a significant increase in consumption among young people.

"However, marketing campaigns geared towards masculinity have resulted in fewer girls consuming energy drinks than boys [3,4]." This is an interesting point. Consider expanding on how these marketing campaigns are geared towards masculinity for clarity.

Methods:

"The study design was a descriptive cross-sectional." Add more details. Why was this design chosen? What advantages does it offer for this research?

"The Tamale Metropolis is an urban area in the Northern Region with a population of about 360,579, 80.8% of which lives in urban areas and a size of 289.58 square miles [15]." This sentence is a bit cluttered. Consider splitting it: "The Tamale Metropolis, located in the Northern Region, has a population of about 360,579. Of this, 80.8% reside in urban areas. The metropolis spans 289.58 square miles [15]."

"The focus of this research was on individuals aged 15 to 45 years old." Why was this age range chosen? Provide a rationale.

Data Collection and Quality Management:

"The study collected data on individual respondents in the Tamale metropolis through semi-structured interviews using a pretested-structured questionnaire administered via face-to-face interviews with the aid of a Computer Assisted Personal Interview (CAPI) device." This is a long sentence. Consider breaking it down for clarity.

"The questionnaire covered demographic information, anthropometry, and perception, with the latter section probing respondents' knowledge of energy drinks." What specific demographic information was collected? Were there any specific scales or metrics used for the perception section.

"Consent was obtained orally. This mode was used to maximize time." Oral consent can be controversial in some research contexts. Elaborate on why this was deemed sufficient and how it was documented.

Results:

"The results of the study indicated that a large percentage of the respondents, 98.7%, had consumed energy drinks before, while 78.7% currently consume them." Consider adding a comparison to other studies or expected outcomes.

"Respondents believed that energy drinks provided additional energy (81.00%) and reduced stress (62.30%). However, they also perceived side effects such as insomnia (60.60%) and restlessness (51.40%)." Consider adding a brief discussion on why these perceptions might exist in your discussion.

"Although the majority of respondents (83.4%) were unaware of the classification of energy drinks and their ingredients, side effects, and benefits." This sentence seems incomplete. Consider completing the thought, e.g., "...the majority were still consuming them regularly."

"Age, marital status, level of education, work intensity, EDs served at gatherings, and knowledge of EDs was significantly associated with ED consumption (p < .05)." Elaborate on each factor's specific impact or correlation strength in your discussion.

Discussion:

"Energy drinks have not only been associated with negative health effects and death but have also been linked to high-risk substance use and social deviance such as vandalism and reckless driving, resulting in accidents [8–11]." This is a strong claim. Ensure that the cited references robustly support this statement.

"Sub-Saharan Africa is experiencing an upward trend in diet-related diseases, including hypertension, obesity, stroke, and heart disease, contributing to the rapid increase in the epidemic of non-communicable diseases." How does this relate to the study's findings? Consider drawing a clearer connection between the broader context and the specific results of this study.

"In Ghana, where energy drink consumption is on the rise, a survey revealed that 62% of student-athletes consume at least one can of energy drink per week." What are the implications of this finding, especially in the context of the study's results.

"The overall aim of this study was to evaluate the youth's perceptions, consumption habits, and factors linked with energy drink consumption in the Tamale Metropolis."

Reiterate the main findings at the beginning of the discussion in relation to this aim to provide a clear summary of the study's contributions.

"This study will make clear enable the understanding of the consumption habits of energy drinks among youth in the Tamale Metropolis will contribute to raising awareness, prompt more research in the area and influence policies and public health actions."

This sentence is convoluted. Rephrase for clarity: "This study enhances our understanding of energy drink consumption habits among the youth in the Tamale Metropolis. The findings can raise awareness, prompt further research, and influence policy and public health initiatives."

6. PLOS authors have the option to publish the peer review history of their article (what does this mean?). If published, this will include your full peer review and any attached files.

Reviewer #1: **Yes: **Dr. Nii Korley Kortei

Reviewer #2: **Yes: **Udoka Okpalauwaekwe

---

## [Author Response · Author response to Decision Letter 0]

3 Dec 2023

We take this opportunity to thank the editor and reviewers of our paper for their kind collaboration to the improvement of this manuscript. We have taken into account all the concerns raised and we have made the suggested modifications. We have implemented numerous improvements to the paper based on these suggestions. Below, we justify our replies to the suggestions made by the respected editor and reviewers of the paper. 

EDITOR

Dear authors, we apologise for the delays in getting you feedback on your paper. We have now secured sufficient peer-review comments on your paper and are happy for you to revise based on their recommendations and submit for further evaluation. This paper is timely given the recent discussions from health and nutrition professionals in Ghana around soft and energy drinks consumption particularly among the youth. However, it is clear from the two reviewers and from my own assessment as Editor that, your paper needs substantial proofreading to tighten up the arguments and correct language errors. Also make sure you back all claims with sufficient evidence.

Thank you for providing us with the feedback from both the reviewers and your own assessment. We appreciate the time and effort invested in reviewing our manuscript. We understand the importance of ensuring the quality and clarity of our work and are committed to addressing the highlighted concerns. Here is our response to the comments. 

a. The paper has undergone substantial proofreading to tighten up arguments and correct language errors.

b. All claims are now backed by sufficient evidence, ensuring a more robust presentation.

1. In addition to comments from the reviewers, consider reducing sections of the discussion and introduce a new section in the discussion on policy measures/recommendations for EDs in Ghana. After reading your findings, a good question is, what next? And so what so we do, in terms of policy or action? Can you link this discussion to the recently released Ghana Food Based Dietary Guidelines? What should be done around schools? What (healthy) alternatives are there and why do you think those will be chosen over energy drinks? 

We have reduced sections of the discussion and introduced a new segment focusing on policy measures and recommendations for energy drinks (EDs) in Ghana. This addition addresses the "what next" aspect and aligns the discussion with the recently released Ghana Food Based Dietary Guidelines. We have explored potential policy implications and actions, particularly in the context of schools, and discussed healthy alternatives to energy drinks. The suggestion titled Recommendations for ED Sale and Consumption is added in the discussion section (see pgs. 20&21, line nos. 532-553). 

"In 2023, Ghana passed a tax bill on sugar sweetened-beverages into law. The law categorizes EDs as sugar sweetened drinks. The policy action is expected to in-crease individual savings and promote health [83]. Taxation is an effective strategy for reducing the consumption of sugar sweetened-beverages. While higher taxes have been more effective (above 10%), such initiatives have no significant impact in high-income countries compared to LMICs [84]. The extant tax regulations exhibit commendable strides in mitigating the consumption of EDs; however, there exists untapped potential for further reduction. It is imperative to reconsider the categorization of EDs given they differ from conventional sugar-sweetened beverages, as they have cardiovascular effects substantiated by several of compelling case studies [11,85–89]. This reclassification stands to facilitate a spectrum of viable interventions, encompassing but not confined to the following: stringent restrictions on the pervasive retail distribution of EDs; the implementation of rigorous age-related restrictions, and prohibition in schools governing the purchase and consumption of EDs; enhanced oversight of marketing, advertising, and product labelling; and the unequivocal disclosure of potential short- and long-term adverse effects associated with the consumption of EDs [6]. 

In alignment with the national food-based dietary guidelines, which stipulate that individual sugar consumption should not surpass 50 grams per day, there is a pressing need for heightened advocacy efforts aimed at fostering decreased sugar intake and curbing the consumption of sugar-sweetened beverages [90]. As EDs are prohibited in school environments, natural fruit drinks should be encouraged. This can be incorporated into the national school feeding program initiative. Additionally, it is essential to provide backing and incentives to beverage manufacturers to invest in the production of healthy, nutrient-rich beverages characterized by low sugar content."

2. What are the limitations of your study? Include this in the discussion. Reviewer 2 highlights some of these in different comments. 

Limitations of the study have been included in the discussion (see pg. 21, line nos. 555-561 ). We have elaborated on the potential shortcomings and challenges faced during the research process, providing a more comprehensive understanding of the study's scope and applicability.

"The adoption of a cross-sectional design restricts the establishment of causal relationships and temporal changes in ED consumption patterns and perceptions. The use of a convenience sample raises concerns about representativeness, potentially limiting the generalizability of findings. Self-report data may introduce recall bias and may not fully reflect actual behaviours and beliefs accurately. The choice of the study area based on personal familiarity introduces potential bias in the selection process. Face-to-face interviews might also introduce interviewer bias and compromise participant anonymity."

 

REVIEWER 1

This is indeed a very good and interesting Public health topic worthy of discussion. There are some suggestions which when added, can boost the strength of this research.

Thank you for your thoughtful and constructive feedback on our research manuscript. We appreciate your positive remarks on the significance of the public health topic. We have carefully considered your suggestions and made the following revisions. 

1. What previous works have been done in Ghana regarding investigation of energy drink consumption? 

We have included a section summarizing previous works on energy drink consumption in Ghana in the introduction section (see pg. 3, line nos. 84-86)., identifying gaps, and highlighting the uniqueness of our study.

"This study builds upon existing research on ED consumption in Ghana, expanding beyond previous investigations that predominantly focused on specific cohorts like students, student-athletes, and drivers [19–24]."

2. What gaps were identified? 

We have included some information in the introduction section on the identified gap (see pg. 3, line nos. 86&88). and highlighting the uniqueness of our study.

"However, they have left a gap in our understanding of ED consumption among the general youth population."

3. How unique is this work or how different is this work from the previous works? 

We have included some information in the introduction section on the identified gap (see pg. 3, line nos. 87&88). and highlighting the uniqueness of our study. 

"This study addresses this gap by examining ED consumption among a representative sample of Ghanaian youth."

4. A map of Ghana showing Tamale and coordinates must be provided. 

A map of Tamale with a label showing the coordinates has been incorporated for better geographic context. We opted to use only the map of Tamale instead of the whole map of Ghana with the map of Tamale highlighted because in that view the Tamale area is not visible enough.

"Fig 1. Map of Tamale (Coordinates: 9.404601, -0.842389)."

5. Last paragraph of the intro must be reconstructed for clarity

“This study will make….. 

The last paragraph of the introduction has been revised for clarity, ensuring a more succinct expression of our study's aims and contributions (see introduction, pg. 4, line nos. 91-93).

"This study enhances our understanding of ED consumption habits among the youth in the Tamale Metropolis. The findings can raise awareness, prompt further research in this domain, and influence policy and public health initiatives."

6. Discussion seems summary of the whole report. Authors need to discuss key findings with sufficient literature 

We have expanded the discussion section to provide more in-depth analysis and interpretation of key findings, incorporating relevant literature to support our arguments. 

(See discussion section, pg. 15, lines 322-329)

"This study found that the popularity of EDs is highly driven by advertising media, especially television. In support of this assertion, Stacey et al. [31] substantiated a correlation between the consumption of EDs and the exposure to promotional advertisements of ED brands. These advertisements were found to typically target a youthful male audience using sports and entertainment-centred themes [32], and leveraging associations with influential celebrities. Gendered advertising reflects the leveraging of higher involvement of males in sports. This is in line with an American study that strongly links ED consumption to traditional masculinity and risk-taking tendencies [33]."

(See discussion section, pg. 16, line nos. 341-343)

"In contrast to this study, prior research conducted in Ghana has yielded dissimilar results, with the most frequently consumed ED brands being identified as Blue Jeans, Lucozade, Rox, Monster, and Red Bull [20,21]."

(See discussion/knowledge section, pg. 16, line nos. 349-361)

"EDs are perceived to have several benefits, including increased alertness, improved work or study concentration, energy boost, stress management, and pro-motion of health and social relationships [15]. This study aligns with previous re-search [22,27,36,37], indicating a prevailing belief in the benefits of EDs, with the highest perceived advantages being increased work or study concentration (20%), energy boost (81%), and stress management (62.3%). Also, this study reported perceived adverse effects including insomnia (60.60%), and restlessness (51.40%) as commonly reported adverse effects of EDs, similar to work by Casuccio et al. [26]. While perception is known to be autonomous concerning thought, knowledge may lead to an intentional intervention in the process of achieving a percept [38]. This suggests perceptions may be influenced by external factors, notably well-targeted advertisements and recommendations from friends and family, which con-tribute to shaping a positive view of EDs among consumers. Understanding these external influences is crucial for devising effective health communication strategies and interventions to promote informed and balanced perspectives on ED consumption."

(See discussion/knowledge section, pg. 16, line nos. 362-370) 

"Further, upon analysing the data, it is evident that a large proportion of the sample population in this study has poor knowledge of EDs, accounting for 83.4%. This trend is consistent with another study that also reported a high rate of poor knowledge among participants [19]. However, in contrast to this study Saku et al., [19] did not find a significant relationship between knowledge and ED use. The findings of this study suggest that low knowledge can result in a high prevalence of ED use. Therefore, it is crucial to address issues of misconceptions and lack of knowledge. For instance, information about the adverse effects of EDs can be incorporated into default marketing strategies. Also, the lack of warning labels and inadequate inclusion of risks and side effects in advertisements and messaging con-tribute to this poor knowledge [19,32,39]."

(See discussion/attitude and practice section, pg. 16&17, line nos. 377-389)

"Moreover, an even higher percentage of individuals have ever tried EDs, with nearly the entire study population (98.7%) reporting consumption at some point. This trend is consistent with studies conducted among adolescents in Shanghai, China (70.5%) and in Ho (85.6%) [19,41]. The high prevalence of ever-consuming EDs is attributed to factors such as curiosity and perceived utility [24,36]. The increasing popularity of EDs is also attributed to various factors, including individual factors such as perceived benefits, targeted marketing strategies, and limited knowledge about these beverages [21,32,42]. Additionally, interpersonal factors like peer pressure, social image, and parental influence also play a role, along with environmental factors such as affordability, availability, and the lack of policies and regulations [41]. In the context of the HBM [30], these external factors serve as triggers of ED consumption. They can significantly impact individuals’ decision to consume EDs. Therefore, such channels of influence must be monitored to ensure that the general public is well-informed to make healthy choices."

(See discussion/attitude and practice section, pg. 17, lines nos. 394-402)

"The main reasons for consuming EDs reported by the respondents were performance enhancement, stress management, energy boost, and increased work or study concentration. In similar studies conducted among drivers, students and athletes, it was reported that EDs were used to manage fatigue, stay awake, and concentrate for extended periods [19,49–51]. Many ED users appreciate their ability to reduce sleep hours, enhance concentration, boost energy, and serve as refreshments. Particularly, workers use EDs to enhance performance in the workplace, while students use them to cope with academic pressure [19,50,51]. Withdrawal symptoms were reported by ED consumers once the effects wore off. This potentially leads to dependence and workplace accidents that is a risk to themselves and others [52,53]."

(See discussion/attitude and practice section, pg. 17&18, lines nos. 416-422)

"EDs are particularly popular among young individuals, especially those in their early twenties [32,36,54]. This can be attributed to the energy-demanding nature of that age group [55]. Young people's curiosity [56] and inclination to follow trends also contribute to their higher consumption of EDs compared to older individuals [43]. Additionally, males generally consume EDs more frequently, particularly at night, and experience more side effects. This aligns with the marketing strategies of EDs, which often target young males and emphasize activities associated with youth and masculinity, such as sports. [6,31,36,54,57,58]."

(See discussion/attitude and practice section, pg. 18, line nos. 424&425) 

"Education level is associated with ED use [26,59]. Accordingly, ED consumption is higher among those with low education levels, and people who work or study long hours [19]."

(See discussion/attitude and practice, pg. 18, line nos. 438-449) 

"According to this study, smokers make up a small proportion of ED consumers. The lack of association between alcohol consumption and ED use is attributed to taste preferences and concerns about the impact of combining the two for performance [62] - one of the primary reasons for consuming EDs among the participants in this study. Similar to previous studies by Casuccio et al. [26]. and Chang et al. [15], no significant associations were found between ED use and these variables. Furthermore, this study revealed that easy access to EDs corresponds to higher consumption rates. EDs are readily available and easily accessible [63] to children and young adults, with numerous brands found in every shop that sells beverages [47,64]. Additionally, there is a lack of government regulations governing the sale and marketing of EDs. This unrestricted promotion and utilization of EDs contribute to their increasing appeal, popularity, and availability [5]. The knowledge or experience of side effects also plays a role in influencing the use of EDs [64]."

(See discussion/predictors of ED consumption, pg. 19, line nos. 497-499)

However, generally, ED consumption decreases with increasing age [40,79,80]. Marital status is a predictor of ED consumption as well [81]. 

7. Make the work aligned to the health belief model

The disc

---

## [Decision Letter · Decision Letter 1]

16 Jan 2024

Title - Energy drinks in Tamale: Understanding youth perceptions, consumption patterns and related factors

PONE-D-23-21713R1

Dear Dr. Kobik,

We’re pleased to inform you that your manuscript has been judged scientifically suitable for publication and will be formally accepted for publication once it meets all outstanding technical requirements.

Kind regards,

Zakari Ali, PhD.

Academic Editor

PLOS ONE

Additional Editor Comments (optional):

Reviewers' comments:

Reviewer's Responses to Questions

**Comments to the Author**

1. If the authors have adequately addressed your comments raised in a previous round of review and you feel that this manuscript is now acceptable for publication, you may indicate that here to bypass the “Comments to the Author” section, enter your conflict of interest statement in the “Confidential to Editor” section, and submit your "Accept" recommendation.

Reviewer #1: All comments have been addressed

Reviewer #2: All comments have been addressed

2. Is the manuscript technically sound, and do the data support the conclusions?

Reviewer #1: Yes

Reviewer #2: Yes

3. Has the statistical analysis been performed appropriately and rigorously? 

Reviewer #1: Yes

Reviewer #2: Yes

4. Have the authors made all data underlying the findings in their manuscript fully available?

Reviewer #1: Yes

Reviewer #2: Yes

5. Is the manuscript presented in an intelligible fashion and written in standard English?

Reviewer #1: Yes

Reviewer #2: Yes

6. Review Comments to the Author

Reviewer #1: The Authors have answered all queries satisfactorily. The work has improved tremendously.

xxxxxxxxxxxxxxxxxxxxxxxxxxxxxxxxxxxxxxxxxxxxxxxxxxxxxxxxxxxxxxxxxxxxx

Reviewer #2: I appreciate the conscientious attention to all the reviewer comments and in providing further context to enhance the readability of your work. Weldone and look forward to reading from you in the further.

7. PLOS authors have the option to publish the peer review history of their article (what does this mean?). If published, this will include your full peer review and any attached files.

Reviewer #1: **Yes: **Nii Korley Kortei

Reviewer #2: **Yes: **Udoka Okpalauwaekwe
